# Epidemiology, Genetic Characterization, and Evolution of *Hunnivirus* Carried by *Rattus norvegicus* and *Rattus tanezumi*: The First Epidemiological Evidence from Southern China

**DOI:** 10.3390/pathogens10060661

**Published:** 2021-05-28

**Authors:** Minyi Zhang, Qiushuang Li, Fei Wu, Zejin Ou, Yongzhi Li, Fangfei You, Qing Chen

**Affiliations:** Guangdong Provincial Key Laboratory of Tropical Disease Research, Department of Epidemiology, School of Public Health, Southern Medical University, Guangzhou 510515, China; myalison@smu.edu.cn (M.Z.); qiushuang@smu.edu.cn (Q.L.); wudiedie1996@smu.edu.cn (F.W.); ouzejin@163.com (Z.O.); Yongzhi_Lee@163.com (Y.L.); fangfei1@icloud.com (F.Y.)

**Keywords:** *Picornavirus*, murine rodents, detection, genome structure, phylogenetic analysis, selective pressure

## Abstract

*Hunnivirus* is a novel member of the family *Picornaviridae*. A single species, *Hunnivirus A*, is currently described. However, there is limited information on the identification of *Hunnivirus* to date, and thereby the circulation of *Hunnivirus* is not fully understood. Thus, the objective of this study was to investigate the prevalence, genomic characteristics, and evolution of rat hunnivirus in southern China. A total of 404 fecal samples were subjected to detection of *Hunnivirus* from urban rats (*Rattus norvegicus* and *Rattus tanezumi*) using PCR assay based on specific primers targeted to partial 3D regions, with the prevalence of 17.8% in *Rattus norvegicus* and 15.6% in *Rattus tanezumi*. An almost full-length rat hunnivirus sequence (RatHuV/YY12/CHN) and the genome structure were acquired in the present study. Phylogenetic analysis of the P1 coding regions suggested the RatHuV/YY12/CHN sequence was found to be within the genotype of *Hunnivirus A4*. The negative selection was further identified based on analysis of non-synonymous to synonymous substitution rates. The present findings suggest that hunniviruses are common in urban rats. Further research is needed for increased surveillance and awareness of potential risks to human health.

## 1. Introduction

*Hunnivirus* belongs to a genus within the family *Picornaviridae*, recently established by the International Committee on Taxonomy of Viruses (ICTV) in 2013 (https://talk.ictvonline.org/ictv/proposals/2013.008a-dV.A.v2.Hunnivirus.pdf, accessed on 3 October 2020). According to the ICTV, the family *Picornaviridae* is currently divided into 68 genera, containing 158 known species (www.picornaviridae.com, accessed on 20 May 2021), many of which result in significant diseases in humans and various animals. *Hunnivirus* is a picornavirus genus including a single species, *Hunnivirus A*. To date, at least nine genotypes of *Hunnivirus* have been described: hunnivirus A1 (formerly bovine hungarovirus 1) [1], hunnivirus A2 (formerly ovine hungarovirus 1) [1], hunnivirus A3 (isolated from sheep cell cultures), hunnivirus A4 (Norway rat hunnivirus) [2], and hunnivirus A5–A9 (https://www.picornaviridae.com/hunnivirus/hunnivirus.htm, accessed on 20 May 2021). By means of genomic sequencing and analysis, the hunniviruses were found to be distinct from kobuvirus when using the primers for kobuvirus screening [1]. The hunnivirus sequences isolated from different host species share a typical genome organization, which are composed of VPg, 5′UTR^IRES-^^II^, a single polyprotein encoding a leader protein (L), P1 (VP4-VP2-VP3-VP1), P2 (2A^npgp^-2B-2C) and P3 (3A-3B^VPg^-3C^pro^-3D^pol^) proteins, 3′UTR, and a poly (A) tail. In addition, the P1 region encodes the viral capsid proteins, while the P2 and P3 regions encode nonstructural proteins related to protein processing and genome replication [1].

In 1965, the identification of the first hunnivirus was detected from sheep cell cultures in Northern Ireland. Subsequently, bovine and ovine hunniviruses were determined in fecal samples of cattle and sheep in central Hungary from 2008 to 2009 [1]. *Hunnivirus* was then proposed as a member of a novel genus in the family *Picornaviridae*. At the request of the ICTV Executive Committee, the name was modified from ‘hungarovirus’ in July 2013 to the name hunnivirus, derived from these two countries, Hungary and Northern Ireland (https://www.picornaviridae.com/hunnivirus/hunnivirus.htm, accessed on 20 May 2021). Between 2014 and 2016, the hunniviruses were identified in commensal Norway rats (*Rattus norvegicus*) in New York City [2] and detected in rodents from several representative regions of China [3]. A novel hunnivirus, provisionally designated feline hunnivirus (FeHuV), has been recently discovered in a cat with diarrhea in Guangdong, China, indicating that pets have the potential to spread members of this genus [4].

In the last few years, the hunniviruses have gradually been identified worldwide in different animals, including cattle, sheep, rats, and cats [1,2,4], which raises interesting questions regarding the global distribution, pathogenesis, and potential risks to human health. However, little is known about these questions, which are thereby challenging to answer. Herein, this study investigated the prevalence, genetic characteristics, and evolution of *Hunnivirus* carried by *Rattus norvegicus* and *Rattus tanezumi* in southern China.

## 2. Results

### 2.1. Detection of Rat Hunnivirus

We investigated 404 fecal samples of rats from Xiamen (n = 31), Malipo (n = 48), Yiyang (n = 83), Guangzhou (n = 146), and Maoming (n = 96) in southern China for the detection of rat hunnivirus. Among these samples, 359 were collected from *Rattus norvegicus* and 45 were collected from *Rattus tanezumi*. Overall, 17.6% (71/404) of specimens were found to be positive for hunnivirus RNA, utilizing the primer set ratHuV-F/R (3D region, 254 nt). *Rattus norvegicus* had a higher positive rate (17.8%, 64/359) of hunnivirus than *Rattus tanezumi* (15.6%, 7/45). Detailed information on sample collection and the prevalence of rat hunnivirus among different locations are shown in Table 1.

### 2.2. Phylogenetic Analysis of Partial 3CD Region Sequences

Twelve representative sequences of the partial 3CD regions (1550 nt) were randomly selected for further PCR amplification and sequencing within the seventy-one positive samples (GenBank accession numbers MW417230–MW417241). Our sequences that were isolated from two different rodent species captured in different geographic locations shared 94.6–99.3% nucleotide identities and 97.2–99.6% amino acid identities with one another. Besides, they displayed the closest relatedness (94.0–95.8% nucleotide and 97.0–99.2% amino acid identities) with a Norway rat hunnivirus sequence (rat/NYC-E21/USA/2012) previously reported in the New York City (accession no. KJ950971.1) [2]. Maximum-likelihood phylogenetic analysis based on the 1550-nt 3CD nucleotide sequences was carried out using other hunnivirus reference sequences in GenBank. The result demonstrates that our sequences clustered along with the rat hunniviruses previously found in Asia and America and formed a tight cluster in a monophyletic branch with a hunnivirus sequence Wencheng-Rt38–2 isolated from *Rattus tanezumi*. These sequences also shared a common root with the hunnivirus sequences identified in felines and rats, but were related more distantly to bovine and ovine hunniviruses (Figure 1).

### 2.3. Genomic Analyses of the Near-Complete Sequence

An almost full-length rat hunnivirus sequence (RatHuV/YY12/CHN, 7282 nt) was successfully obtained from *Rattus norvegicus* (accession no. MW417242). This sequence included a partial 5′ untranslated region (UTR) of 481 nt, one complete ORF of 6705 nt that encoded a potential polyprotein of 2234 amino acids, and a partial 3′ UTR of 96 nt. Further, we predicted the genome organization and potential cleavage sites for our near-complete sequence. It was found that the L protein was 210 nt (70 aa) in length, whereas the complete P1, P2, and P3 regions were 2337 (779 aa), 1827 (609 aa), and 2331 nt (776 aa) long, respectively. The detailed genome structure of RatHuV/YY12/CHN is shown in Figure 2A. In addition, we presented the predicted RNA secondary structure of 5′UTR by RNAfold, evidencing that the 481-nt 5′UTR sequence can construct a stable secondary structure with the minimum free energy of −164.80 kcal/mol (Figure 3).

The findings of BLASTn analysis indicated that our complete polyprotein sequence shared the highest nucleotide and amino acid identities with the rat/NYC-E21/USA/2012 sequence, with values of 91.2 and 92.7%, respectively. In contrast to other rat-derived hunnivirus sequences, RatHuV/YY12/CHN shared 70.3–79.5% nucleotide identities and 71.5–79.1% putative amino acid identities with two Vietnamese sequences and one Chinese sequence. Interestingly, RatHuV/YY12/CHN had a higher sequence similarity with the FeHuV identified in Guangdong, China, at the nucleotide (85.1%) and putative amino acid (86.5%) levels. However, it shared a relatively low nucleotide/amino acid identity with bovine (67.6/70.1%) and ovine (67.7/69.9%) hunnivirus sequences. The similarities of each functional region were further compared between RatHuV/YY12/CHN and other reported hunniviruses in GenBank (Table 2). We found the highest nucleotide (97.2%) and amino acid (97.4%) identities presented in the rat/NYC-E21/USA/2012 L and P2 regions, respectively. In comparison, the lowest nucleotide (63.0%) and amino acid identities (45.9%) occurred in the P2 and L regions of bovine hunnivirus sequence, respectively. Overall, RatHuV/YY12/CHN shared 61.2–92.9%, 63.5–93.1%, 70.4–97.4%, and 81.2–96.9% amino acid identities with the rat-derived hunniviruses at the L, P1, P2, and P3 regions, respectively. Taken together, the genetic diversity is existent in the hunniviruses from different host species. Additionally, a similarity plot analysis was conducted to further analyze the genetic characteristics of our near-complete nucleotide sequence, with the bovine hunnivirus sequence BHUV1/HUN/2008/JQ941880.1 used as the out-group sequence and the Norway rat hunnivirus sequence rat/NYC-E21/USA/2012/KJ950971.1 used as the query sequence. The findings indicate that RatHuV/YY12/CHN exhibited relatively high similarities to the query sequence in the 5′ UTR, L, VP4, VP3, 2B-2C, and from the 3A to 3D regions, while different similarities were presented in the VP2, VP1, and 2A regions, and the sequence connecting 2B and 2C. Moreover, the P1 coding region was referred to as having a higher range of genetic variability (Figure 2B).

### 2.4. Phylogenetic Analyses and Negative Selection during the Evolution of Hunnivirus

The maximum-likelihood phylogenetic tree was constructed to investigate the phylogenetic relationship between RatHuV/YY12/CHN and other *Hunnivirus A* strains based on the complete P1 coding regions. The analytical results demonstrated that RatHuV/YY12/CHN clustered with the Chinese FeHuV sequence (accession no. MF953886.1), within the genotype of *Hunnivirus A4* (Figure 4). These P1 sequences in the phylogenetic tree were then aligned for selective pressure analyses. Despite no positively selected sites, the strong negative selection on the non-synonymous sites was detected in SLAC, FEL, and REL methods [5], with the negatively selected sites of 319, 519, and 340, respectively.

To further characterize the evolution of *Hunnivirus* and detect possible differences in selective pressure between phylogenetic branches, we obtained the annotated complete coding regions (CDSs) in the hunnivirus reference sequences in GenBank. Then, we reconstructed the phylogenetic tree using the synonymous sites (Figure 5). The analytical findings revealed that our sequence RatHuV/YY12/CHN clustered into a single group with the Chinese feline and rat hunnivirus sequences. The mean *dN/dS* values were calculated using the GA Branch method for each branch, indicating that all the phylogenetic branches were under strong negative selection (0 < *dN/dS* < 1).

## 3. Discussion

In 1965, the hunnivirus was first discovered in sheep cell cultures. However, it was not reported in cattle, sheep, rats, and cats until the last few years [1,2,4]. The detailed data related to hunnivirus in different host species and different geographic locations are quite limited. To the best of our knowledge, this study represents the first study to specifically investigate the prevalence of hunniviruses in fecal samples from *Rattus norvegicus* and *Rattus tanezumi* in southern China. We investigated 404 fecal samples of which 71 (17.6%) were positive for hunnivirus using self-designed primers. The mean detection rates of *Hunnivirus* in *Rattus norvegicus* and *Rattus tanezumi* were 17.8 and 15.6%, respectively, similar to the prevalence of rat hunnivirus (16%, 21/133) in the USA and bovine hunnivirus 1 (15%, 4/26) in central Hungary [1,2]. Nevertheless, information on the global distribution of *Hunniviruses* remains to be determined due to the lack of epidemiological evidence. Further studies are warranted to investigate the geographic distribution and host species of *Hunnivirus* worldwide, thereby confirming variations in prevalence among different host species.

In previous studies, the hunniviruses were serendipitously identified when detecting other picornaviruses using the primers UNIV-kobu-F/UNIV-kobu-R. These primers were designed to screen three prototypes of kobuviruses (Aichi virus, bovine kobuvirus, and porcine kobuvirus) based on the 3D conserved viral regions [6]. To date, numerous kobuviruses have been determined worldwide among various host species using this primer pair, including human [7], cattle [8,9], pig [10], cat [11,12], dog [13], deer [14], and wolf [15]. Similarly, an epidemiological study of rat kobuvirus isolated from murine rodents in Guangdong, China, was previously reported by our research team [16]. However, this primer pair was considered more specific for picornaviruses than for kobuviruses [1,6], confirming the universal applicability in identifying novel picornaviruses [4]. For instance, the bovine hunnivirus was initially identified in a cattle sample but further found in three additional samples after using specific hungarovirus screening primers (Hungaro-3D-R/F) [1]. Likewise, our present study found that one 215-nt-long nucleotide sequence had no similarity to kobuviruses available in the GenBank database but shared 83.2–87.9% nucleotide identity with five hunnivirus sequences by BLAST.

To address this issue, we designed generic hunnivirus screening primers (RHuV-F/RHuV-R) based on the 3D conserved region to determine rat hunniviruses from additional fecal samples and consequently, expected PCR products of amplicon size (254 nt) for hunnivirus were identified. We confirmed the low sensitivity of the primers UNIV-kobu-F/UNIV-kobu-R that may reflect the detection rate of hunnivirus in rats [1]. Additionally, we also screened kobuvirus for hunnivirus-positive specimens, of which 15 specimens were co-infected with kobuvirus. This finding was consistent with a prior study demonstrating that kobuviruses frequently mixed infections with other pathogens [17]. Hence, further investigation should be carried out to determine whether the co-detection or multiple virus detection is equally common for hunniviruses and other pathogens.

Maximum-likelihood phylogenetic analysis based on the partial 3CD regions suggested that the nucleotide sequences of rat kobuvirus isolated from *Rattus norvegicus* and *Rattus tanezumi* clustered tightly together in a clade according to the species, but regardless of being derived from different geographical locations (Figure 1). Moreover, our sequences formed a group with other rat-derived hunnivirus sequences identified in China, Vietnam, and the USA. Interestingly, they also clustered with a Chinese FeHuV (accession no. MF953886.2), and the majority shared relatively high nucleotide (95.1–96.6%) and amino acid identities (97.0–99.2%) with the Chinese FeHuV sequence [4]. What is more, the phylogenetic relationships based on P1 regions and complete coding regions between our sequence and other hunnivirus sequences (*Hunnivirus A1–A9*) illustrated that our rat hunnivirus sequence shared a common root with the Chinese FeHuV sequence (Figure 4 and Figure 5). The combined findings suggest that the cross-species transmission of hunnivirus possibly occurs in rats and other animal species, which would result in a sustained threat to public health due to the increased contact between humans and these animals. Since less attention has been focused on the potential risk of cross-species transmission for hunniviruses, it is of significance to conduct serological investigations and assess the pathogenicity posed to human health.

The almost complete gene sequence of RatHuV/YY12/CHN (7282nt) was successfully acquired in the present study. Except for the 5′ and 3′ UTR regions, the genetic analyses suggested that the polyprotein structure of RatHuV/YY12/CHN could be cleaved into 12 viral proteins, including L, P1 (VP4, VP2, VP3, and VP1), P2 (2A–2C), and P3 (3A–3D). The results of a comparative sequence analysis for each functional region show that RatHuV/YY12/CHN had a high amino acid identity with FeHuV in the complete poly-protein gene and different functional regions, especially in the P3 region (97%). Interestingly, the P1 structural protein showed fewer nucleotide and amino acid identities (mostly < 70%) between RatHuV/YY12/CHN and other hunnivirus reference sequences, demonstrating that genetic diversity is exhibited and the P1 region might be the most variable motif in the *Hunnivirus* genome (Table 2). Such highly variable motifs encoding the significant viral capsid protein might determine the pathogenicity and antigenicity for picornavirus [18]. Thereby, we used SLAC/FEL/REL methods in the HyPhy package when calculating the *dN/dS* values at each P1 sequence site, identifying strong negative selection on the non-synonymous sites. Taking SLAC results as an example, a total of 319 negatively selected sites and no positively selected site were reported with a statistical significance of 0.05. Additionally, the mean *dN/dS* ratio (0.0170) represented the average 98.3% of non-synonymous mutations which were eliminated by negative selection during the evolution of virus.

A branch-specific analysis using the HyPhy package (GA Branch analysis) was subsequently generated because the *dN/dS* ratio between branches in the phylogenetic tree may vary when hunnivirus sequences in the alignment come from different host species (Figure 5). The findings demonstrated evidence of complex and variable selective pressures on the hunnivirus sequences, with the strong support of negative selection along all branches representing the evolution in each host species. Therefore, the negative selection might be responsible for the evolution of P1 and even the entire coding regions of hunniviruses, which deserve putatively functional studies in the future.

This study predicted the positions of protease-cleavage sites in the rat hunnivirus viral proteins by sequence alignment with known cleavage sites in hunnivirus sequences, which contained E/G, A/D, Q/G, P/T, and G/P (Figure 2A). The cleavage sites between VP4 and VP2 (A/D), 2A and 2B (G/P), 3B and 3C (Q/G), and 3C and 3D (Q/G) are conserved among the feline, bovine, ovine, and several rat-derived hunniviruses, in line with a recent study reporting in detail on the protease-cleavage sites of hunniviruses [4]. We considered that understanding and feature analysis concerning the protein cleavage sites would be helpful for the in-depth investigations into the mechanism of protein cleavage [19]. Furthermore, recent studies have illustrated that functional non-coding RNAs have critical impacts on defects functionally related to various diseases [20,21]. Several studies have revealed the predicted RNA secondary structure of picornaviruses, while little is known currently on this structure in hunniviruses. In this study, the analysis of predicted RNA secondary structure showed that the 5′UTR region of our sequence can construct a stable secondary structure with the minimum free energy of −164.80kcal/mol (Figure 3). Since the functions of non-coding RNAs are highly associated with their structures, understanding the structures could clarify the functions of non-coding RNAs. Thus, more 5′ UTR sequence data should be obtained to determine the genetic features of *Hunnivirus*.

The number of recently determined gastrointestinal viruses is expanding rapidly. To date, many picornaviruses are associated with diarrhea among humans and other animals. A previous review summarized the viruses that have been described in gastroenteritis cases by next-generation sequencing platforms, such as human *Aichi virus* [22,23], *Cosavirus* [24], and *Salivirus* [25,26,27]. The evidence combined with the epidemiological studies and statistical analysis showed that kobuviruses might be a potential causative agent for diarrhea disease in calves and pigs [28,29]. Although a feline hunnivirus described previously was found in a cat with diarrhea, the ability of hunniviruses to cause diarrhea disease remains to be determined. Therefore, identifying hunniviruses from different host species with and without diarrhea is needed to explore the potential association between the prevalence of *Hunnivirus* and diarrhea disease in the future.

## 4. Materials and Methods

### 4.1. Sample Collection

A total of 404 fresh fecal samples were collected between October 2015 and September 2017 from rats with unknown health status captured close to human residences using live traps in five different areas in southern China, including Xiamen in Fujian Province, Malipo in Yunnan Province, Yiyang in Hunan Province, and Guangzhou and Maoming in Guangdong Province (Figure 6). Individual fresh stools were immediately placed in RNase-free tubes with 700 μL phosphate-buffered saline (PBS; 0.3% homogenate) and kept frozen at −80 °C until further use.

### 4.2. Nucleic Acid Extraction and cDNA Synthesis

The thawed stool specimens were fully resuspended in PBS and centrifuged at 8000× *g* for 10 min at 4 °C to collect the supernatants. Following the manufacturer’s instructions, viral nucleic acid was extracted from 200 μL of each supernatant using the MiniBEST Viral RNA/DNA Extraction Kit (TaKaRa, Kusatsu, Japan). The obtained RNA was reverse transcribed to synthesize cDNA using a Transcriptor First Strand cDNA Synthesis Kit (Roche, Basel, Switzerland). The cDNA was directly used as the template for polymerase chain reaction (PCR) or stored at −20 °C.

### 4.3. PCR Detection for Hunnivirus

During the present study, a hunnivirus sequence was serendipitously found in the fecal sample. Thus, we designed a pair of specific primers for rat hunnivirus to determine additional hunniviruses from the collected fecal samples (RHuV-F: 5′-TGGTGACCGGACTGATGGACCC-3′, corresponding to nucleotides [nt] 6291–6312 of the sequence rat/NYC-E21/USA/2012 and RHuV-R: 5′- TCAGTTCAGCATGCAGCACCGG-3′, corresponding to 6523–6544 nts of rat/NYC-E21/USA/2012), targeting a 254-bp fragment at the partial 3D gene. Among the 3D gene-positive samples, a pair of nucleotide primers (RHuV-3CD-F/R) were subsequently used to amplify a longer partial 3CD region (Table 3). The PCR conditions were 94 °C for 3 min, 40 cycles of 94 °C for 30 s, 56 °C for 1 min, 72 °C for 1 min, with a final extension at 72 °C for 10 min. Amplicons were subjected to 1.0% agarose gel electrophoresis and sequenced using an ABI Prism 3730xl DNA Analyzer (Applied Biosystems, Foster City, CA, USA).

### 4.4. Near-Complete Genome Amplification

Nine pairs of primers were designed to amplify the near-complete genome of rat hunnivirus using the Benchling website (https://benchling.com, accessed on 3 May 2020), based on the reference sequences in GenBank (accession no. KJ950971.1 and MF352430.1). The primer sequences are shown in Table 3. After sequencing, one almost full-length rat hunnivirus sequence was assembled using Lasergene SeqMan software (DNASTAR, Inc. Madison, WI, USA).

### 4.5. Genetic and Phylogenetic Analyses

The nucleotide sequences of rat hunniviruses identified in the current study were compared to the corresponding sequences of other hunnivirus sequences available in GenBank by Basic Local Alignment Search Tool (BLAST; https://www.ncbi.nlm.nih.gov, accessed on 22 July 2020). The pairwise nucleotide and amino acid identities among all sequences were calculated in the MegAlign program (DNASTAR, Inc., Madison, WI, USA). Multiple sequence alignment was carried out by using the CLUSTAL W program in Molecular Evolutionary Genetics Analysis (MEGA version 6.0, Oxford Molecular Ltd., New York, UK). The open reading frame (ORF) was predicted for the obtained sequence utilizing ORF finder (https://www.ncbi.nlm.nih.gov/orffinder, accessed on 15 September 2020). RNA secondary structure prediction was generated using RNAfold 2.4.13 (http://rna.tbi.univie.ac.at/, accessed on 23 September 2020). Phylogenetic tree construction was generated by the maximum-likelihood method with 1000 bootstrap replications in MEGA v6.0 and visualized in FigTree v1.4.0 (http://tree.bio.ed.ac.uk/software/figtree/, accessed on 8 September 2020). The model selections were based on the results of “Find best DNA/Protein models” in MEGA v6.0. A similarity plot analysis of the almost full-length genome was conducted using SimPlot 3.5.1 software.

### 4.6. Selective Pressure Analyses

The signatures of selection from homologous sequences were inferred by estimating the relative rates of non-synonymous (*dN*) and synonymous (*dS*) substitutions. To detect site-specific selective pressure on the codon alignment of P1 regions, we implemented the HyPhy package on the Datamonkey Webserver (www.datamonkey.org, accessed on 25 May 2021) [5] using three codon-based methods. The methods included single-likelihood ancestor counting (SLAC), fixed effects likelihood (FEL), and random effects likelihood (REL). Further, the genetic algorithm (GA) Branch analysis was used to vary selection pressure between different phylogenetic branches using the non-synonymous and synonymous substitutions ratio (*dN/dS*). It noted that the sequence alignments came from translated protein sequences and then mapped aligned residues to codons to avoid introducing frameshifts and preserve codons.

### 4.7. Data Summary

The partial 3CD nucleotide sequences and near-complete genome of rat hunnivirus sequence have been lodged within the GenBank database under accession numbers MW417230 to MW417242.

## 5. Conclusions

In conclusion, the present study is expected to be the first to provide an epidemiological manifestation for rat hunnivirus from *Rattus norvegicus* and *Rattus tanezumi* in southern China, indicating that hunniviruses are common in rats close to humans’ residences. Our findings from genetic analyses suggest that genetic diversity is displayed among the hunniviruses from different host species. Furthermore, the almost complete genome of RatHuV/YY12/CHN was successfully sequenced in this study. The phylogenetic trees based on the P1 coding regions demonstrate that the RatHuV/YY12/CHN belongs to the genotype of *Hunnivirus A4*. Strong negative selection was detected based on non-synonymous to synonymous substitution rates, providing novel insights into the viral evolution. Even though the list of bioinformatical analyses performed in the current study is supposed to be incomplete, the present results will be helpful to understand the limitation of the epidemic, molecular characteristics, and evolution of hunnivirus in rats in China. These provide an excellent beginning to dive even deeper into viral analysis. Therefore, there is a high need for research focused on the epidemiology, geographic distribution, heterogeneity, pathogenesis, transmission route, and the potential risk of cross-species transmission of rat hunnivirus.

## Figures and Tables

**Figure 1 pathogens-10-00661-f001:**
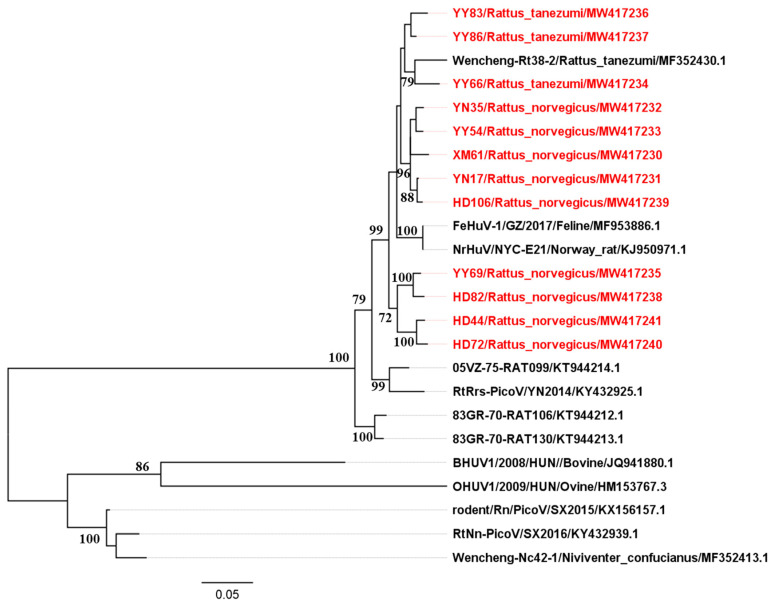
Phylogenetic tree of *Hunniviruses* based on the partial 3CD nucleotide sequences. The tree was generated by the maximum-likelihood method based on the General Time Reversible (GTR) model (gamma-distributed with invariant sites (G + I) and partial deletion) with 1000 bootstrap replicates, and the statistics values > 70% are displayed above the tree branches. Red font represents the sequences of rat hunniviruses identified in the present study.

**Figure 2 pathogens-10-00661-f002:**
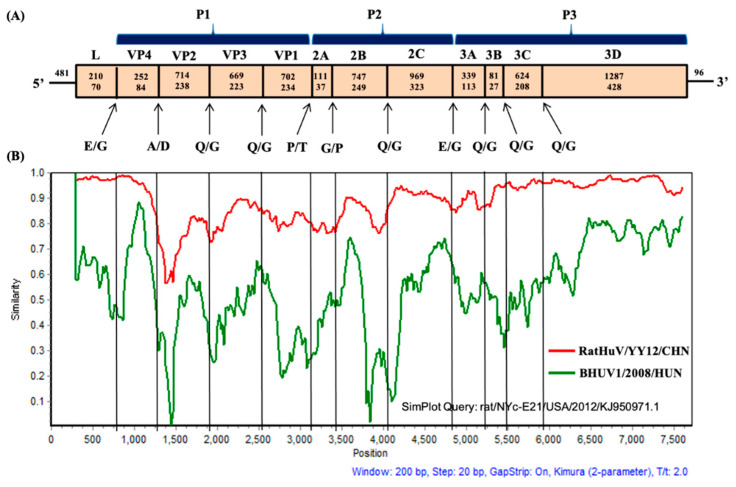
Genome organization and genetic characterization of the RatHuV/YY12/CHN. (**A**) Graphical depiction of the almost complete genome organization of RatHuV/YY12/CHN, including each functional region (*above bar*) and the predicted cleavage sites (*below bar*) determined by the multiple alignments with other *Hunnivirus* reference sequences and nucleotide positions. The 5′ and 3′ UTRs and the open reading frame (*box*) are shown. P1 represents viral structural proteins, and P2 and P3 represent nonstructural proteins. Nucleotide (*upper number*) and amino acid (*lower number*) lengths are displayed in each region box. (**B**) Similarity plot analysis of the near-complete genome of RatHuV/YY12/CHN (red line). The bovine hunnivirus BHUV1/HUN/2008/JQ941880.1 (green line) was used as an out-group sequence, and Norway rat hunnivirus rat/NYC-E21/USA/2012/KJ950971.1 as a query sequence using Kimura (2-parameter) model in Simplot 3.5.1 software.

**Figure 3 pathogens-10-00661-f003:**
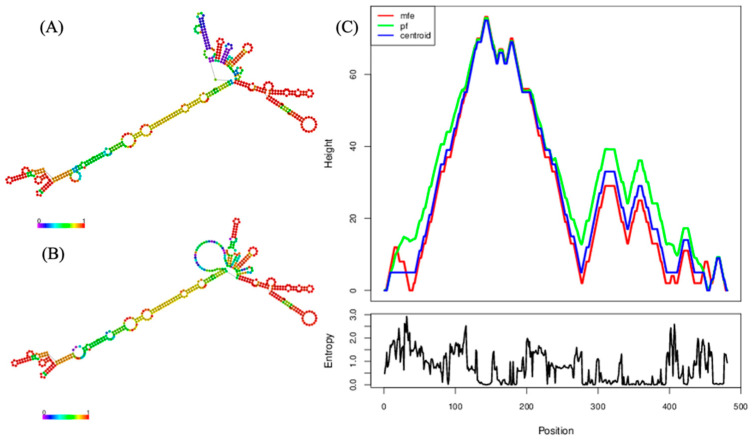
Predicted RNA secondary structure of the rat hunnivirus sequence RatHuV/YY12/CHN 5′ UTR obtained with RNAfold. (**A**) Minimum free energy predicted structure (-164.80kcal/mol). (**B**) Centroid secondary structure. Base-pairing probabilities color both images. For unpaired regions, the color denotes the probability of being unpaired. (**C**) Mountain plot of minimum free energy structure, the thermodynamic ensemble of RNA structures, the centroid structure, and positional entropy for each position.

**Figure 4 pathogens-10-00661-f004:**
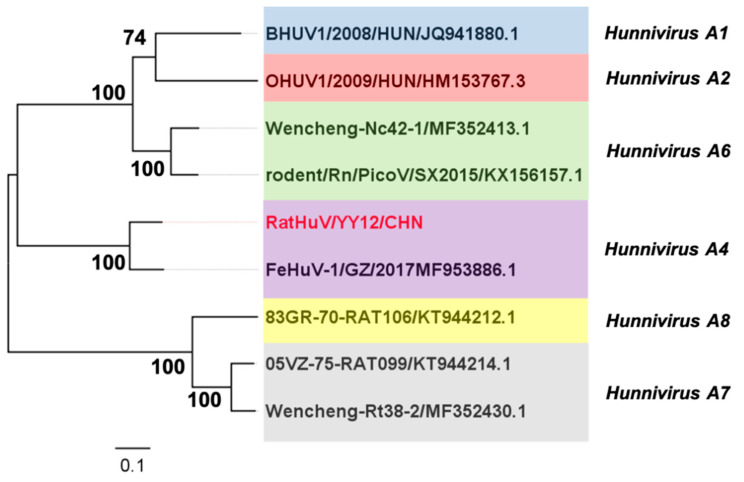
Phylogenetic tree of *Hunniviruses* based upon the P1 coding regions. The tree was generated by the maximum-likelihood method based on the General Time Reversible (GTR) model (gamma-distributed with invariant sites (G + I) and partial deletion) with 1000 bootstrap replicates. The red font represents the sequence of rat hunnivirus identified in the present study.

**Figure 5 pathogens-10-00661-f005:**
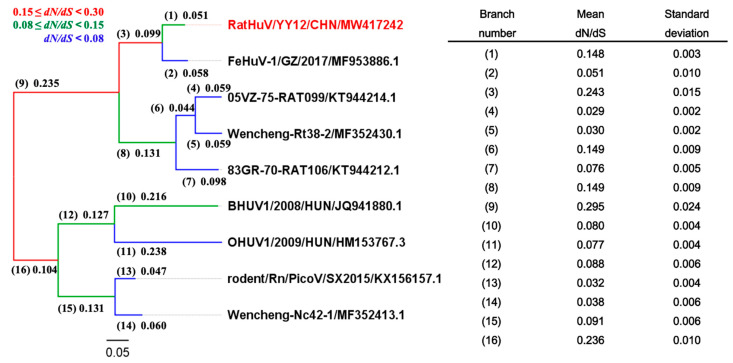
Branch-specific analysis of selection along the phylogenetic tree based on complete CDSs of *Hunnivirus*. Analysis was conducted using the GA Branch method from the HyPhy package on the Datamonkey webserver. Left: the phylogenetic tree of RatHuV/YY12/CHN and other reference hunniviruses was generated by the maximum-likelihood method based on the General Time Reversible (GTR) model (gamma-distributed with invariant sites (G + I) and partial deletion) with 1000 bootstrap replicates. The red font represents the sequence of rat hunnivirus identified in the present study. The branch number (in the parenthesis) and branch length (*dS*) are presented. The color code for each branch class is displayed on top. Right: the estimated mean *dN/dS* values are indicated ± standard deviation.

**Figure 6 pathogens-10-00661-f006:**
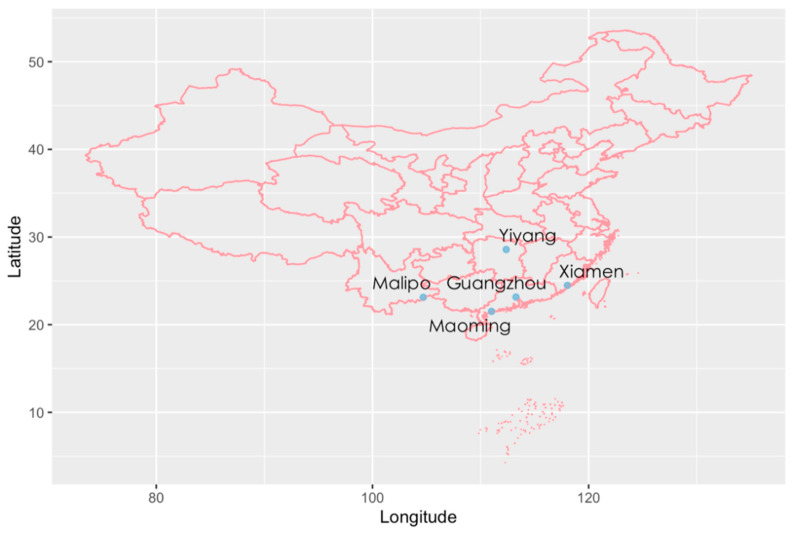
Specimen locations for hunnivirus in southern China. The distribution map of fecal samples collected from five representative cities of southern China was constructed by R software version 3.6.3 in the *maps* and *mapdata* packages. The locations are labeled with blue points.

**Table 1 pathogens-10-00661-t001:** Prevalence of hunnivirus in *Rattus norvegicus* and *Rattus tanezumi* from five locations of China.

Region	*Rattus Norvegicus*		*Rattus Tanezumi*		Total
	Number of Samples	Prevalence	Number of Samples	Prevalence	
**Xiamen**	18	5.6% (1/18)	13	-	3.2% (1/31)
**Malipo**	47	21.3% (10/47)	1	100% (1/1)	22.9% (11/48)
**Yiyang**	68	27.9% (19/68)	15	33.3% (5/15)	28.9% (24/83)
**Guangzhou**	139	17.3% (24/139)	7	14.3% (1/7)	17.1% (25/146)
**Maoming**	87	11.5% (10/87)	9	-	10.4% (10/96)
**Total**	359	17.8% (64/359)	45	15.6% (7/45)	17.6% (71/404)

**Table 2 pathogens-10-00661-t002:** Nucleotide and putative amino acid sequence identity of the complete polyprotein gene and L, P1–P3 regions between RatHuV/YY12/CHN and other *Hunnivirus* reference sequences ^a^.

Gene Region	Rat				Feline	Bovine	Ovine
	Rat/NYc-E21/USA/2012	83GR-70-RAT106/Vietnam/2012	05VZ-75-RAT099/Vietnam/2013	Rat/Wencheng-Nc42–1/China/2012	FeHuV-1/GZ/2017	BHUV1/HUN/2008	OHUV1/HUN/2009
L	97.2/92.9	94.5/88.2	95.3/88.2	79.9/61.2	97.2/91.8	63.8/45.9	74.0/56.5
P1	85.2/93.1	61.8/63.5	61.3/63.5	65.3/70.3	85.2/84.6	65.2/69.9	63.4/68.0
P2	90.7/97.4	80.4/87.6	80.0/89.3	67.5/70.4	90.7/86.7	63.0/66.8	64.3/67.5
P3	94.9/96.9	91.0/94.1	91.3/95.1	76.6/81.2	94.9/97.0	73.8/80.5	73.9/80.6
Polyprotein	91.2/92.7	79.5/78.3	78.7/79.1	70.3/71.5	85.1/86.5	67.6/70.1	67.7/69.9

Values described as nucleotide identity (%)/amino acid identity (%). ^a^ Reference *Hunnivirus* strains include rat/NYc-E21/USA/2012 (accession no. KJ950971.1), 83GR-70-RAT106/Vietnam/2012 (accession no. KT944212.1), 05VZ-75-RAT099/Vietnam/2013 (accession no. KT944214.1), rat/Wencheng-Nc42–1/China/2012 (accession no. MF352413.1), FeHuV-1/GZ/2017 (accession no. MF953886.1), BHUV1/HUN/2008 (accession no. JQ941880.1), and OHUV1/HUN/2009 (accession no. HM153767.3).

**Table 3 pathogens-10-00661-t003:** Information for primer sequences.

Reaction Number	Primer Name	Primer Sequences (5′-3′)	Size of PCR Products (bp)
1	UNIV-kobu-F	TGGAYTACAAG(/R)TGTTTTGATGC	216
	UNIV-kobu-R	ATGTTGTTRATGATGGTGTTGA	
2	RHuV-F	TGGTGACCGGACTGATGGACCC	254
	RHuV-R	TCAGTTCAGCATGCAGCACCGG	
3	RHuV-3CD-F	GGATATTTYCCCCGCGGCAARG	1610
	RHuV-3CD-R	ATAGTCTTGC TCCCCGCGGTGT	
4	RHuV-S-F1	AGTGACCCCATGCGAAGTGCTG	1044
	RHuV-S-F2	CCCTTGTGTGTCTGAGCGCCAC	899
	RHuV-837-R	GACGAGTCGCCACCTCCAGCAC	
5	RHuV-335-F	ATCTGGGGCCCTGTCTGGAGTG	1191
	RHuV-1508-R	GAGTCCAAGGGGCACTCTGGGT	
6	RHuV-1204-F1	TAAGAACACACCAGCTGCCGCG	1701
	RHuV-2171-F2	GCAAATCCCAGCAGTGGTGGCT	734
	RHuV-2883-R	GCCCCCATGGTGTTGAGTTGGG	
7	RHuV-2659-F	CTACCCACCAGGTTCACACGTA	1970
	RHuV-4607-R	CTGTCTTGTAGGCGGCACCAGC	
8	RHuV-3647-F	GACTTGAGCAACTTTGGACCAA	1584
	RHuV-5207-R	CTCTCTTGCCGGCRGARTTRCC	
9	RHuV-4608-F	CTGGTGCCGCCTACAAGACAGC	1148
	RHuV-5734-R	CCCAGGTGCTGTTTGGGCTCTG	
10	RHuV-5192-F	CAGGGTGCCTATGGCGGTAACT	1138
	RHuV-6308-R	TCAGTTCAGCATGCAGCACCGG	
11	RHuV-6194-F	GGGACTGACAACCTGGACCCGA	984
	RHuV-7156-R	ATAGTCTTGCTCCCCGCGGTGT	
12	RHuV-6526-F	ATCCGCCGTTGGGACAAATCCG	866
	RHuV-E-F	TGCTCTGGGGAAAAATAACCCT	

R: A/G.

## Data Availability

All data generated or analyzed during this study are included in this published article. Access to raw data can be acquired by contacting the corresponding author via email.

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
