# Peer review of "Epidemiology, Genetic Characterization, and Evolution of Hunnivirus Carried by Rattus norvegicus and Rattus tanezumi: The First Epidemiological Evidence from Southern China"

_pathogens, 2021, doi:10.3390/pathogens10060661_

Round 1

Reviewer 1 Report

Table 1 title. This table is about prevalence in the two rat species. A table title that reflects that succinctly would be: Prevalence of hunnivirus in Rattus norveigicus and R. tanezumi from five regions of China.

Line 193-194. Needs some clarified language. Substitute ”similar to” for “accordance with which” prevalences for …

Lines 222-224 do you mean occurrence rather than detection?

The discussion needs better focus. Paragraphs 1 and 2 of the discussion belong in the introduction. The remaining paragraphs need to start with a simple, declarative lead sentence that states what the paragraph is about or statement of a conclusion followed by sentences that support the statement. The last and concluding paragraph should start with sentence for a clear call for research to establish the health importance of Hunniviruses.

Author Response

Responses to Reviewer 1 comments

Many thanks for your useful comments and suggestions on our manuscript. These comments and suggestions are of vital benefit to enhance the quality of the manuscript and our future work. We have made some changes and tried our best to improve the manuscript. Please see our point-by-point responses as listed below.

Point 1: Table 1 title. This table is about prevalence in the two rat species. A table title that reflects that succinctly would be: Prevalence of hunnivirus in Rattus norveigicus and R. tanezumi from five regions of China.

Response 1: Thanks for your suggestion. We have removed “Sample collection and information on rat hunnivirus detection” and changed it to “Prevalence of hunnivirus in Rattus norvegicus and Rattus tanezumi from five regions of China” in the revised manuscript. (Page 2, Line 108)

Point 2: Line 193-194. Needs some clarified language. Substitute “similar to” for “accordance with which” prevalences for …

Response 2: The words “in accordance with which” were rewritten to “similar to” in the revised manuscript. (Page 7, Line 564)

Point 3: Lines 222-224 do you mean occurrence rather than detection?

Response 3: We are sorry for the confusing description. We mean detection here and we have rewritten this sentence in the revised manuscript. (Page 7, Line 590-591)

Point 4: The discussion needs better focus. Paragraphs 1 and 2 of the discussion belong in the introduction. The remaining paragraphs need to start with a simple, declarative lead sentence that states what the paragraph is about or statement of a conclusion followed by sentences that support the statement. The last and concluding paragraph should start with sentence for a clear call for research to establish the health importance of Hunniviruses.

Response 4: Many thanks for your suggestion regarding the discussion section. We have improved the statement throughout the revised discussion as suggested. (Page 7-8)

Reviewer 2 Report

This paper reports on the prevalence and genetic diversity of hunninviruses in China, as sampled from 2 rat species.  The paper is scientifically robust, with appropriate methodology, analyses and inferences.  The paper would have had more impact and interest if it had sampled other species likely to come into contact with these rats, but the paper is fine as it is.

Comments

Editing and proof reading – some words missing, would be good to improve English expression throughout the manuscript with the aid of an editor (if possible)

First paragraph – aka is colloquial, spell out the acronym

Line 39 – this line is confusing as there is only one species of hunnivirus

Line 67, 69, 70 and throughout the manuscript – italics for species names

Line 69 – where does the primer set come from?

Figure 6 seems low quality; could it be improved?

Methods: for the SLAC, REL etc. analyses it's not clear what the dataset is and the number of sequences.  Can this be provided as an appendix?

The RNAfold analysis seems standard but doesn't really have a purpose.  What does it tell us about hunniviruses?  I felt this was not discussed adequately. 

Author Response

Responses to Reviewer 2 comments

Many thanks for your useful comments and suggestions on our manuscript. These comments and suggestions are of vital benefit to enhance the quality of the manuscript and our future work. We have made some changes and tried our best to improve the manuscript. Please see our point-by-point responses as listed below.

Point 1: Editing and proof reading – some words missing, would be good to improve English expression throughout the manuscript with the aid of an editor (if possible)

Response 1: Many thanks for your comment. We have tried our best to improve the English expression throughout the revised manuscript.

Point 2: First paragraph-aka is colloquial, spell out the acronym

Response 2: We are sorry for the incorrect word used in this paragraph. The word “aka” was removed from the revised manuscript. (Page 1, Line 36)

Point 3: Line 39 – this line is confusing as there is only one species of hunnivirus.

Response 3: We are sorry for the confusing presentation. We have rewritten the sentence in the revised manuscript. (Page 1, Line 39-40)

Point 4: Line 67, 69, 70 and throughout the manuscript – italics for species names

Response 4: We have double-checked throughout the manuscript and revised the species names in italics. (Page 2, Line 97, 103-106)

Point 5: Line 69 – where does the primer set come from?

Response 5: The primer set was designed by our research groups. We have included how it design in the methodology section (Page 9, Line 1010-1028).

Point 6: Figure 6 seems low quality; could it be improved?

Response 6: Thank you for your suggestion. We have improved the quality of Figure 6 in the revised manuscript. (Page 9, Line 996)

Point 7: Methods: for the SLAC, REL etc. analyses it's not clear what the dataset is and the number of sequences. Can this be provided as an appendix?

Response 7: We are sorry for the unclear information. For SLAC, REL etc. analyses, we used the same P1 sequences as those in the phylogenetic tree, as shown in Figure 4 (Page 6, Line 522). Similarly, the complete coding regions of our sequence and hunnivirus reference sequences as shown in Figure 5 were used to perform GA-Branches analysis (Page 6, Line 535). There were both 9 hunnivirus sequences used in the analyses because not much hunnivirus sequences were currently available in the GenBank. The accession numbers for these reference sequences were also included in the Figures. Therefore, we consider that the dataset and the number of sequences could be descripted in the result section, so we have added relevant information in the revised manuscript. (Page 5, Line 349-350; Page 6, 527-530)

Point 8: The RNAfold analysis seems standard but doesn't really have a purpose.  What does it tell us about hunniviruses?  I felt this was not discussed adequately.

Response 8: We are sorry that we did not discuss it adequately. We have included relevant information on RNAfold analysis in the revised manuscript. (Page 8, Line 824-833)